# Distributed forward Brillouin sensor based on local light phase recovery

Desmond M. Chow [1], Zhisheng Yang [1], Marcelo A. Soto [1,2] & Luc Thévenaz [1]

The distributed fibre sensing technology based on backward stimulated Brillouin scattering (BSBS) is experiencing a rapid development. However, all reported implementations of distributed Brillouin fibre sensors until today are restricted to detecting physical parameters inside the fibre core. On the contrary, forward stimulated Brillouin scattering (FSBS), due to its resonating transverse acoustic waves, is being studied recently to facilitate innovative detections in the fibre surroundings, opening sensing domains that are impossible with BSBS. Nevertheless, due to the co-propagating behaviour of the pump and scattered lights, it is a challenge to position-resolve FSBS information along a fibre. Here we show a distributed FSBS analysis based on recovering the FSBS induced phase change of the propagating light waves. A spatial resolution of 15 m is achieved over a length of 730 m and the local acoustic impedances of water and ethanol in a 30 m-long uncoated fibre segment are measured, agreeing well with the standard values.

---

[1] EPFL Swiss Federal Institute of Technology, Institute of Electrical Engineering, SCI-STI-LT Station 11, CH-1015 Lausanne, Switzerland. [2] Present address: Department of Electronic Engineering, Universidad Técnica Federico Santa María, 2390123 Valparaíso, Chile. Correspondence and requests for materials should be addressed to D.M.C. (email: desmond.chow@epfl.ch)

Backward stimulated Brillouin scattering (BSBS), commonly known as stimulated Brillouin scattering (SBS), is widely known for enabling distributed fibre sensing[1,2]. While the spatial resolution, number of resolved points and real-time sampling rate have improved greatly over the past few years, distributed Brillouin sensing is nonetheless restricted to strain and temperature measurements or their direct physical derivatives due to the confinement of longitudinal acoustic wave inside the fibre core, which is an inherent limit in diversifying the options for detected parameters. On the other hand, forward SBS (FSBS), also known as guided acoustic-wave Brillouin scattering (GAWBS)[3], is an opto-acoustic effect occurring in optical fibres in which electrostriction-induced transverse acoustic eigenmodes are stimulated by intense optical waves propagating in the fibre core. Because of the cross-sectional finite dimension, the acoustic waves resonate transversally inside the fibre. This intriguing acoustic wave has been exploited for various innovative acoustic-based metrologies, namely for fibre diameter estimation[4], sound velocity measurement[5], strain[6], temperature[7,8] sensing, and more recently for measuring the acoustic impedance of surrounding liquid materials[9]. The latter sensing application allows the mechanical properties of an external material to directly impact the detected optical signal. Currently, the reported optical detection schemes rely on either a Sagnac interferometer loop configuration to convert nonreciprocal optical phase variations of a probing signal into changes in the optical intensity for polarised (radial) FSBS[10] or a direct polarised light detection method for depolarised (torsional) FSBS[3]. However, both techniques result in measurements that are not spatially resolved[9,11]. Since FSBS scatters light in forward direction only because of the phase matching condition[12,13], neither backscattering-based time-of-flight acquisition[14] nor local correlation points[15,16] can be realised directly with the FSBS scattered light, making the measurement of its local resonance spectrum a challenging and non-trivial process.

In this paper, we demonstrate a technique to measure the distributed FSBS resonance spectrum of an optical fibre for the first time, to the best of our knowledge. The principle of our technique is based on measuring the longitudinal phase evolution of a guided light that is perturbed by FSBS transverse acoustic waves. First, a long optical pulse, on which an intensity modulation is superimposed at the FSBS resonant frequency (in the radio frequency (RF) range), is used for stimulating a transverse acoustic wave in an optical fibre. Then, a reading optical pulse that is in another wavelength range and follows the FSBS activating pulse will be phase modulated by the transverse acoustic wave, thus generating multiple sidebands on the reading pulse spectrum. As the amplitude of the oscillating phase changes experienced by the reading pulse accumulates along the fibre, the intensity of each sideband changes, following the corresponding order of Bessel functions. By analysing the intensities of the sidebands, the phase changes experienced by the reading pulse can be retrieved.

Since the reading pulse sidebands are narrowly-spaced and forward propagating, a specific technique is needed to selectively measure their intensity progressions. Considering the narrow resonance linewidth and large frequency-detuning features of the BSBS process, the intensity evolution of several reading pulse sidebands are independently selected and measured as a function of distance over a sensing fibre; this particular step is similar to Brillouin optical time-domain analysis (BOTDA)[1,2]. Both transversal and longitudinal acoustic waves (activated by FSBS and BSBS, respectively) are independently and distinctively used in our technique without cross-interaction. Subsequently, the longitudinal phase evolution of the reading pulse is retrieved mathematically from the measured intensity progressions of several

reading pulse sidebands. Finally, the phase evolution is numerically differentiated with respect to the longitudinal distance to obtain the local response of FSBS. The local FSBS resonance spectrum is then reconstructed by repeating the steps while detuning the FSBS activating frequency around the FSBS resonance peak. In this work, distributed measurements of the FSBS resonance spectrum of a 730 m single-mode fibre (SMF) are achieved with a spatial resolution of 15 m.

The transverse acoustic wave resonating inside the optical fibre experiences a partial reflection at the optical fibre boundary due to the acoustic impedance mismatch between the surroundings and the fibre material (silica). This reflection traps the transverse acoustic wave within the cross-sectional round cavity, affecting the decay time of the acoustic field[9] as well as the spectral linewidth of the FSBS resonances[17]. For sensing demonstration, the local resonance linewidth in a 30 m uncoated fibre segment is retrieved and the acoustic impedances of water and ethanol are locally calculated, corresponding well with the reported standard values. The demonstrated distributed FSBS analysis is not limited to sensing applications, but also aiming at verifying position-dependent FSBS theoretical models[18] and characterising local FSBS in a waveguide, which are as well within the scope of this analysis technique.

## Results

**Activation of the FSBS transverse acoustic wave.** The cylindrical structure of an optical fibre supports discrete sets of longitudinal, radial, torsional, and flexural elastic vibrations that could be stimulated through electrostriction. The elastic waves present in the forward light scatterings are the radial modes and the mixed torsional-radial modes, which define acoustic waves that propagate radially and circumferentially, respectively. Only the pure radial transverse acoustic waves, referred simply as transverse acoustic waves hereafter, are considered in this work because they are reflected normally from the surface of the fibre cladding, thus providing the strongest response yet the simplest case for the acoustic impedance analysis. Due to the bounded fibre cross-section, the transverse acoustic waves are confined into eigenmodes. The intensity of the induced transverse acoustic eigenmodes varies across the acoustic spectrum because the modes that have larger overlaps with the optical field will be activated more efficiently. One of the strongest induced transverse acoustic modes, which corresponds to the 7th mode with resonant frequency $\nu_{res} = 322$ MHz in standard SMF, is used for the sensing demonstration in this work. For further details regarding the guiding properties of the transverse acoustic resonances in standard optical fibres, see Supplementary Note 1, Supplementary Figures 1 & 2 and Supplementary Table 1.

Via FSBS, two coherent light waves of frequencies $\omega_1$ and $\omega_2$, respectively, are coupled with the transverse acoustic resonance of frequency $\Omega = \Delta\omega = \omega_1 - \omega_2$. Momentum conservation requires that the acoustic and optical wave vectors $\mathbf{K}(\Omega)$ and $\mathbf{k}(\omega_m)$, respectively, satisfy the phase matching relation, $\mathbf{K}(\Omega) = \Delta\mathbf{k} = \mathbf{k}(\omega_1) - \mathbf{k}(\omega_2)$. In the case of forward light scatterings, the group velocity of the activating light $v_g$ is equivalent to the axial phase velocity of the transverse acoustic wave $V_a$, $v_g = \Delta\omega/\Delta|\mathbf{k}| = \Omega/|\mathbf{K}| = V_a$. Due to the negligible group velocities of the transverse acoustic eigenmodes for small axial wave vectors, the same acoustic resonant frequencies maintain for a broad range of axial phase velocities, thus relaxing the phase matching condition and allowing the transverse acoustic wave to be stimulated by light at any telecom wavelength. The FSBS activating light is created by intensity modulating an incoming light with sinusoidal signal, forming a spectrum with sidebands separated by the

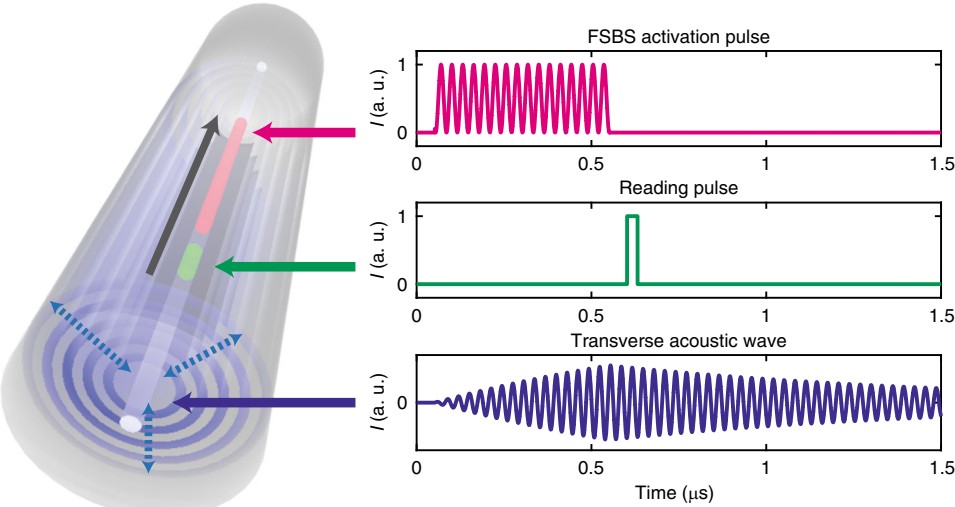

**Fig. 1** Activation and reading of the transverse acoustic wave at an arbitrary fibre position. An intensity modulated optical pulse (magenta) stimulates a transverse acoustic wave (blue) (the oscillating period has been enlarged for illustration purposes), whose spectrum is probed by the reading pulse (green) that is time-overlapping with a short segment of the acoustic tail. The reading pulse is phase modulated by the transverse acoustic wave, thus generating multiple spectral sidebands. The longitudinal evolution of each reading pulse sideband resulting from this phase modulation is independently mapped by the process of backward stimulated Brillouin scattering. FSBS forward stimulated Brillouin scattering

modulation frequency $\nu_F$. The beating among these spectral components inside the optical fibre stimulates the transverse acoustic mode that has a resonant frequency corresponding to $\nu_F$. The intensity modulated light that activates FSBS is shaped into a long rectangular pulse to time-isolate the Kerr cross-phase modulation (discussed in the next section). This pulse is represented by the magenta trace in Fig. 1 and is referred to as FSBS activating pulse hereafter.

At an arbitrary fibre position $z$, the activated transverse acoustic wave remains oscillating locally for a certain decay time after the FSBS activating pulse passes through, as shown in the blue curve of Fig. 1. Through photoelastic effect, the transverse acoustic wave periodically modifies the refractive index of the fibre core at the modulation frequency $\nu_F$. Considering negligible fibre loss, the peak power of the FSBS activating pulse is constant along the fibre, thus the induced refractive index change can be expressed as:

$$\Delta n(\nu_F, z, t) = a_{RI}(\nu_F, z) \cos(2\pi \nu_F t) \qquad (1)$$

where $a_{RI}(\nu_F, z)$ is the amplitude of the refractive index changes and represents the spectrum of the selected FSBS resonance mode at position $z$, from which the local acoustic impedances of the external materials are eventually retrieved.

The experimental configuration related to the FSBS activating process is illustrated in the magenta path shown in Fig. 2. The continuous wave (CW) light from a distributed feedback (DFB) laser at 1533 nm is intensity modulated with sinusoidal signal at the selected FSBS resonant frequency ($\nu_{res} = 322$ MHz) using an electro-optic modulator (EOM), followed by shaping the modulated light into a ~500 ns rectangular pulse using another EOM. Approximately 160 periods of the intensity modulation are contained within this long pulse. The FSBS activating pulse is amplified by a high power erbium-doped fibre amplifier (EDFA) to reach a peak power of ~35 dBm so that the generated transverse acoustic wave is sufficiently strong for the subsequent reading process. To obtain the spectral profile of $a_{RI}(\nu_F, z)$, the modulation frequency $\nu_F$ is detuned around the FSBS resonant frequency $\nu_{res}$ and is expressed as $\nu_F = \nu_{res} + \Delta\nu_F$ (the scanned frequency detuning $\Delta\nu_F$ is from $-7$ MHz up to 7 MHz, resulting

in $\nu_F$ ranging from 315 MHz up to 329 MHz with an incremental step of 0.1 MHz).

**Reading the FSBS-induced phase modulation**. The induced refractive index changes represented by Eq. 1 can be probed with a guided probing light that propagates in the same direction as the FSBS activation pulse but in a different wavelength region, so that the FSBS activating and reading (probing) processes are optically isolated. To provide time-of-flight distributed measurements of the FSBS-induced refractive index changes, this probing light is shaped into a short pulse (see green trace in Fig. 1), and is mentioned as the reading pulse. The optical field of the reading pulse can be expressed as:

$$E(z_r, t) = A(z_r, t) \exp\left[j(k_z z_r - \omega_0 t + \varphi_0 + \Delta\varphi)\right] \qquad (2)$$

where $\varphi_0$ is the initial phase, $\Delta\varphi$ is the phase shift due to the transverse acoustic wave, $A(z_r, t)$ is the spatio-temporal profile of the reading pulse, $\omega_0$ is the optical carrier frequency, $z_r$ is the position of the reading pulse in the fibre, $k_z$ is the axial optical wave-vector $k_z = 2\pi/\lambda$, $n_{eff}$ is the effective refractive index and $\lambda$ is the optical wavelength. Since the longitudinally phase velocity of the transverse acoustic wave is equal to the group velocity of the guided light, this reading pulse will follow the same acoustic phase along the fibre, therefore each temporal point of the reading pulse experiences the induced refractive index change associated to the respective overlapping point with the transverse acoustic waveform. In other words, the transverse acoustic waveform is imprinted on the reading pulse as phase shifts $\Delta\varphi$. This scenario is equivalent to the reading pulse being phase modulated by the transverse acoustic wave inside the fibre. Considering that the reading pulse duration is much longer than the period of the FSBS phase modulation, the phase shift $\Delta\varphi$ is given by

$$\Delta\varphi(\nu_F, z_r, t) = \frac{2\pi}{\lambda} \int_0^{z_r} \Delta n(\nu_F, z, t) dz = \frac{2\pi}{\lambda} \cos(2\pi \nu_F t) \int_0^{z_r} a_{RI}(\nu_F, z) dz \qquad (3)$$

The reading pulse undergoes a phase shift $\Delta\varphi$ that accumulates over distance as it propagates along the fibre. For simplicity, we

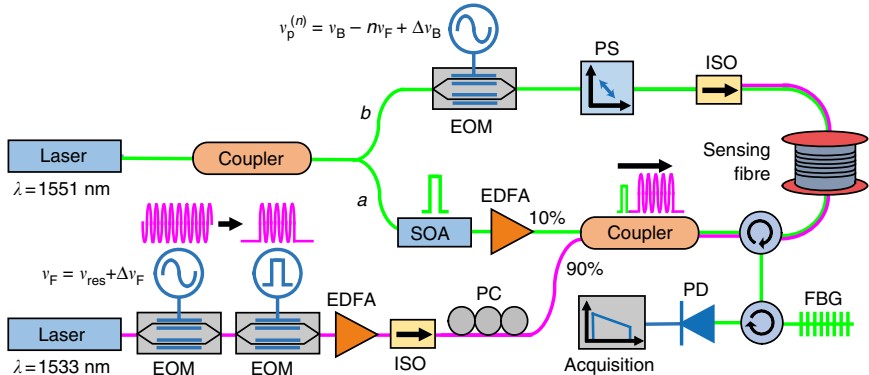

**Fig. 2** Experimental setup for the distributed measurement of forward stimulated Brillouin scattering. Schematic diagram of the apparatus used to measure the spatially resolved spectrum of forward stimulated Brillouin scattering (FSBS). The FSBS activation light is represented as the magenta path ($\lambda = 1533$ nm) and the probing system based on backward stimulated Brillouin scattering (BSBS) is represented as the green path ($\lambda = 1551$ nm). The sensing fibre is 730 m-long and consists of a 30 m uncoated segment spliced between two coated segments of 500 m and 200 m respectively. The components are labelled as follows: EOM electro-optic modulator, ISO isolator, PC polarisation controller, EDFA erbium-doped fibre amplifier, SOA semiconductor optical amplifier, PS polarisation switch, PD photodetector, FBG fibre Bragg grating. The mathematical symbols are defined as follows: $\nu_{res}$, the selected FSBS resonant frequency, $\nu_F$, frequency of the intensity modulation and $\Delta\nu_F$, scanned frequency detuning for FSBS activation; $\nu_B$, Brillouin frequency shift of BSBS; $\nu_p^{(n)}$, frequency of the intensity modulation and $\Delta\nu_B$, scanned frequency detuning for BSBS probe

define a term called phase factor

$$\xi(\nu_F, z_r) = \frac{2\pi}{\lambda} \int_0^{z_r} a_{RI}(\nu_F, z) dz \qquad (4)$$

which represents the amplitude of the integrated phase shift experienced by the reading pulse as it reaches the position $z_r$ in the fibre. Thus, the amplitude of local refractive index change $a_{RI}(\nu_F, z)$ can be recovered through differentiation. By combining Eqs. 2–4, the optical field can be written as

$$E(\nu_F, z_r, t) = A(z_r, t) \exp\left[j\left(k_z z_r - \omega_0 t + \varphi_0\right)\right]$$
$$\exp\left[j\xi(\nu_F, z_r) \cos(2\pi\nu_F t)\right] \qquad (5)$$

The second exponential term of Eq. 5 can be expanded via Jacobi-Anger identity and results in:

$$E(\nu_F, z_r, t) = A(z_r, t) \exp\left[j\left(k_z z_r - \omega_0 t + \varphi_0\right)\right]$$
$$\left[\sum_{n=-\infty}^{\infty} j^n J_n(\xi(\nu_F, z_r)) \exp(jn\Omega_F t)\right]$$
$$= A(z_r, t) \exp\left[j\left(k_z z_r + \varphi_0\right)\right] \qquad (6)$$
$$\left[\sum_{n=-\infty}^{\infty} j^n J_n(\xi(\nu_F, z_r)) \exp(-j(\omega_0 - n\Omega_F)t)\right]$$

where $\Omega_F = 2\pi\nu_F$. Eq. 6 indicates that the spectrum of the reading pulse consists of spectral sidebands that are offset from the optical carrier frequency to multiples of the modulation frequency $\omega_0 \pm n\Omega_F$. The intensity progression associated to the $n$th order spectral sideband $I_r^{(n)} = E_{(n)}E_{(n)}^*$, is proportional to the square of the ordinary Bessel function of order $n$, $J_n^2(\xi)$ (see Fig. 3). By knowing the local intensities of the reading pulse sidebands, the phase factor $\xi(\nu_F, z_r)$ can be calculated mathematically (discussed further in the phase factor recovery and data processing section).

In the experiment, the intense FSBS activating pulse simultaneously generates an undesired phase modulation in the reading pulse wavelength due to Kerr cross-phase modulation, which has the same waveform signature but in phase quadrature. When both FSBS and Kerr cross-phase modulation present together, the detected spectrum shows a Fano resonance shape that complicates the retrieval of the acoustic resonance linewidth[19,20]. Knowing the contrast between the instantaneous characteristic of Kerr response and the slow decay of transverse acoustic wave (typically lasting about 1 μs in an uncoated fibre[9]), the impact of Kerr effect on the measurement is avoided by delaying the reading pulse after the FSBS activating pulse. This way, the reading pulse interacts only with the tail of the FSBS transverse acoustic wave (blue trace in Fig. 1), which is free from the Kerr-induced distortions. The experimental setup related to the reading pulse is shown in the green path $a$ in Fig. 2. Light from a DFB laser at 1551 nm is split into two paths to create counter-propagating pump and probe lights for BOTDA measurements (discussed in the next section). The reading pulse, which is also the pump pulse for the BOTDA system, is created by shaping the incoming light into a 30 ns high-extinction ratio pulse using a semiconductor optical amplifier (SOA). The reading pulse is then injected into the sensing fibre through a 90:10 coupler and synchronised by instruments to propagate with a 10 ns delay after the FSBS activating pulse.

**Intensity progression of the reading pulse sidebands**. To obtain the longitudinal phase evolution, the intensity progressions $I_r^{(n)}(z_r)$ of a few reading pulse sidebands have to be individually measured (see the next section for further details). BSBS is exploited for this measurement since the linewidth of the Brillouin gain spectrum (BGS) (~30 MHz) is much narrower than the separation between reading pulse sidebands $\nu_F$. Through BSBS, the pump wave transfers energy proportional to its intensity to a counter-propagating probe wave that is red-shifted by the Brillouin frequency shift (BFS) ($\nu_B \approx 10.86$ GHz in standard optical fibre). As the frequency difference between one of the reading pulse sidebands and the probe wave is equal to the BFS, the sideband under interrogation will map its intensity variation efficiently to the probe wave during propagation in the fibre[21,22].

The BSBS measurement technique used here to retrieve $I_r^{(n)}(z_r)$ is similar to BOTDA (see the Methods section). The individual sideband of the reading pulse acts as the pump pulse for the BOTDA system whereas the CW probe wave is created by intensity modulating the incoming light with a sinusoidal signal at frequency $\nu_p^{(n)} = \nu_B - n\nu_F$ using an EOM (see green path $b$ of

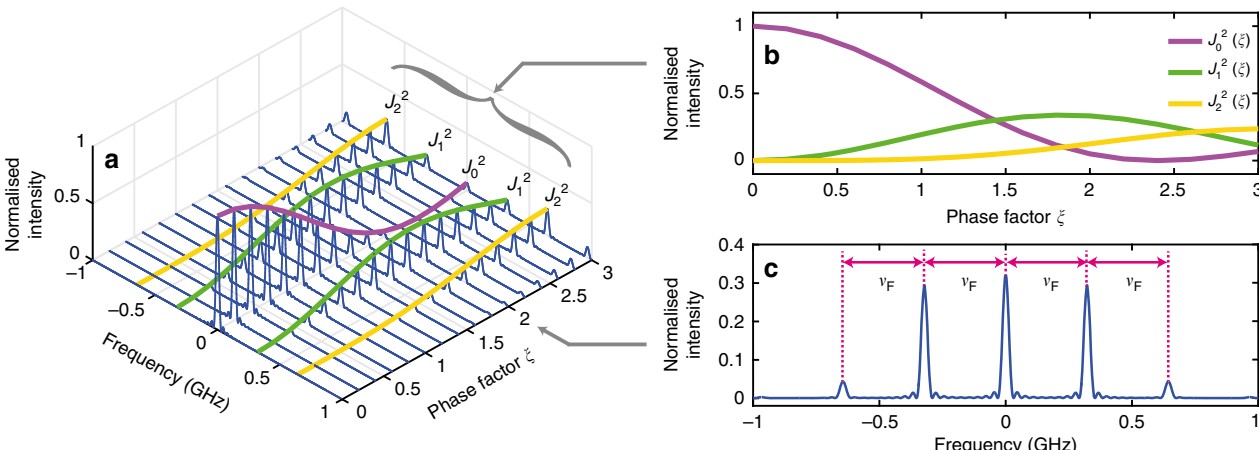

**Fig. 3** Simulated longitudinal evolution of the reading pulse spectrum. **a** The reading pulse spectrum changes along the fibre as it is being phase modulated by the transverse acoustic wave of forward stimulated Brillouin scattering (FSBS). (For illustration purpose, we assume that the fibre surroundings is uniform along the fibre, i.e., the phase factor $\xi(\nu_F, z_r)$ increases linearly with respect to the reading pulse location $z_r$. The spectrum is normalised by the input intensity of the reading pulse at its central frequency). **b** The intensity progression of the 0th, 1st and 2nd order spectral sidebands (each one proportional to $J_0^2$, $J_1^2$ and $J_2^2$, respectively) of the reading pulse as a function of the $z_r$-dependent phase factor $\xi(\nu_F, z_r)$. **c** The reading pulse spectrum at an arbitrary position. Note that the local spectral sidebands are separated by the modulation frequency of FSBS activation

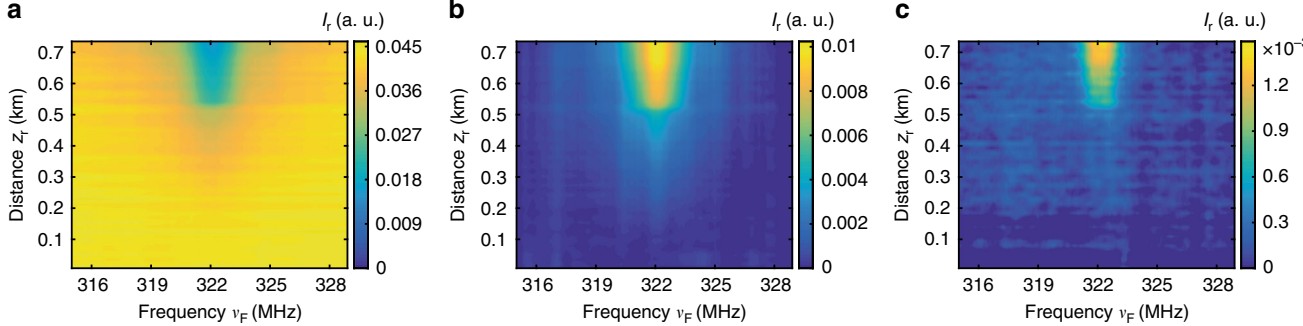

**Fig. 4** Measured intensity progressions of the reading pulse sidebands. Intensity $I_r^{(n)}(\nu_F, z_r)$ of the **a** 0th, **b** 1st, and **c** 2nd order reading pulse sidebands (each one is proportional to $J_0^2$, $J_1^2$, and $J_2^2$, respectively), measured at each scanning frequency $\nu_F$ (detuned within $\Delta\nu_F$ around $\nu_{res}$). Experimental results are obtained when the 30 m uncoated fibre segment is exposed to air

Fig. 2), so that the probe frequency coincides with the BGS peak of the $n$th order reading pulse sideband. Since the CW probe wave and the FSBS activating pulse are counter-propagating, the refractive index perturbation induced on the probe wave averages to zero, thus the probe wave is independent from the FSBS activation process. Nevertheless, the BFS in reality is not uniform along the fibre due to the inherent material inhomogeneity, external temperature perturbations and strain variations, which would affect the precise measurement of the sideband intensity progression. To overcome this problem, the peak gain of the BGS has to be determined by detuning $\nu_p^{(n)}$ around the BFS, $\nu_p^{(n)} = \nu_B - n\nu_F + \Delta\nu_B$ (the frequency detuning $\Delta\nu_B$ is from $-25$ to $25$ MHz). Then, a quadratic fitting is applied to the local BGS to obtain the peak gain, which proportionally represents the local intensity of the interrogated reading pulse sideband. Here, the BGS scanning to obtain $I_r^{(0)}(z_r)$ is repeated for all detuned frequencies $\Delta\nu_F$ around the FSBS resonant frequency $\nu_{res}$ and again repeated for other two higher-order sidebands, $I_r^{(1)}(z_r)$ and $I_r^{(2)}(z_r)$; the results are shown in Fig. 4.

**Phase factor recovery and data processing.** The intensity progression of $n$th order reading pulse spectral sideband $I_r^{(n)}(z_r)$ is a function of phase factor $\xi(\nu_F, z_r)$ and is proportional to $J_n^2(\xi(\nu_F, z_r))$, written as $I_r^{(n)}(z_r) = I_r^{(n)}(\xi(\nu_F, z_r)) = \kappa J_n^2(\xi(\nu_F, z_r))$. In principle, to retrieve $\xi(\nu_F, z_r)$, the inverse function of $I_r^{(n)}(\xi(\nu_F, z_r))$ has to be determined. Although the non-invertible functions $J_n^2(\xi(\nu_F, z_r))$ can be made invertible by limiting the domain and range to the regions that are injective, using a single sideband to measure $\xi(\nu_F, z_r)$ is nevertheless unreliable because the longitudinal intensity progression can be easily affected by external perturbations, such as fibre bending or local losses. A more robust approach, which avoids the detrimental impact of local losses, is using the recurrence properties of Bessel functions to recover $\xi(\nu_F, z_r)$. In particular, a simple form of the Bessel recurrence relation[23] is given as

$$J_{(n-1)}(\xi(\nu_F, z_r)) + J_{(n+1)}(\xi(\nu_F, z_r)) = \frac{2n}{\xi(\nu_F, z_r)} J_n(\xi(\nu_F, z_r)) \quad (7)$$

where $n$ is the Bessel function order. Using Eq. 7, $\xi(\nu_F, z_r)$ can be reliably determined from the intensities of three consecutive orders of spectral sidebands. Here, we select the 0th, 1st, and 2nd order sidebands to maximise the signal-to-noise ratio (SNR) of the measurements as they have the highest intensities as compared to the other higher-order sidebands. $\xi(\nu_F, z_r)$ is obtained from Eq. 7 with $n = 1$, and is related to the measured intensities

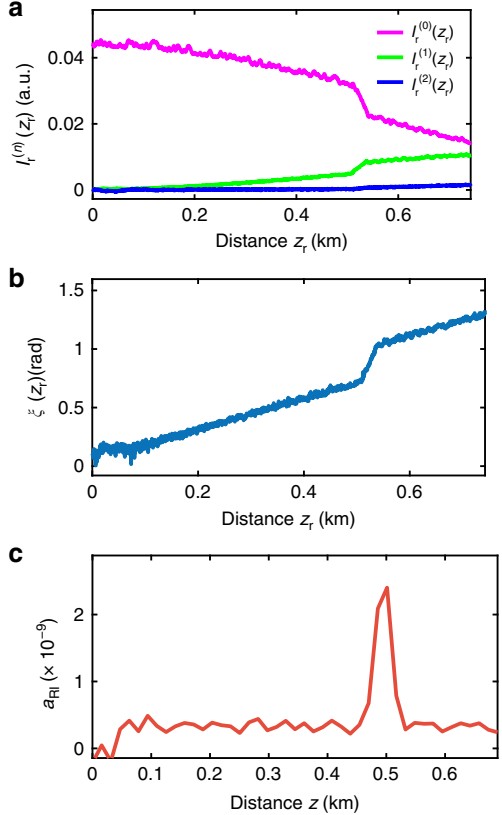

**Fig. 5** Data processing steps to obtain the local response of forward stimulated Brillouin scattering. **a** The measured intensity $I_r^{(n)}(z_r)$ of the 0th, 1st, and 2nd order spectral sidebands versus distance $z_r$, obtained at the FSBS peak resonant frequency $\nu_{res}$. **b** Phase factor $\xi(z_r)$ retrieved by using the Bessel recurrence relation in Eq. 8. **c** The local amplitude of refractive index changes $a_{RI}(z)$ (unitless) that is obtained by using segmental differentiation

of the sidebands $I_r^{(n)}(\xi(\nu_F, z_r))$, as given below:

$$\xi(\nu_F, z_r) = 2\left(\frac{J_1(\xi(\nu_F, z_r))}{J_0(\xi(\nu_F, z_r)) + J_2(\xi(\nu_F, z_r))}\right)$$
$$= 2\left(\frac{\sqrt{I_r^{(1)}(\xi(\nu_F, z_r))}}{\sqrt{I_r^{(0)}(\xi(\nu_F, z_r))} + \sqrt{I_r^{(2)}(\xi(\nu_F, z_r))}}\right) \text{for } 0 \leq \xi(\nu_F, z_r) \leq 2.4048$$

(8)

Note that all $I_r^{(n)}(\xi(\nu_F, z_r))$ in the numerator and denominator of Eq. 8 are of the same degree, thus any power loss affecting the reading pulse, which scales all $I_r^{(n)}(\xi(\nu_F, z_r))$ equally, does not change $\xi(\nu_F, z_r)$. Since Eq. 8 involves taking the square root of $I_r^{(n)}(\xi(\nu_F, z_r))$, the signs of $J_n(\xi(\nu_F, z_r))$ cannot be recovered, which restricts the equation's validity to the regions where $J_0(\xi(\nu_F, z_r))$, $J_1(\xi(\nu_F, z_r))$, and $J_2(\xi(\nu_F, z_r))$ are positives. To overcome this problem, we limit the operation to the first valid region, which begins at $\xi(\nu_F, z_r) = 0$ and ends at the first zero of $J_0(\xi(\nu_F, z_r))$, i.e., $\xi(\nu_F, z_r) = 2.4048$. In practice, the intensity of the FSBS activating pulse, which affects the phase factor $\xi(\nu_F, z_r)$, is controlled carefully so that $I_r^{(0)}(\xi(\nu_F, z_r))$ does not exceed the first minimum at the end of the fibre and $\xi(\nu_F, z_r)$ remains in the valid region. Nevertheless, it must be mentioned that limiting $\xi(\nu_F, z_r)$ is a simplified solution for demonstration purpose. Extending the calculation to all regions is possible as $\xi(\nu_F, z_r)$ is a monotonic function, which implies the changes of signs for $J_n(\xi(\nu_F, z_r))$ follow

a definite sequence, thus allowing $\xi(\nu_F, z_r)$ to be retrieved successively along the distance according to the progress of $\xi(\nu_F, z_r)$.

The measurement results of $I_r^{(n)}(z_r)$ as shown in Fig. 4 are stored in a computer and processed trace by trace for each scanning frequency $\nu_F$ within the measurement range. To describe the data processing steps, only the traces measured at the FSBS resonant frequency $\nu_{res}$ are detailed. $I_r^{(n)}(z_r)$ of the 0th, 1st, and 2nd order spectral sidebands are plotted in Fig. 5a. $\xi(z_r)$ is obtained by applying Eq. 8 pointwise at each location $z_r$, the result is shown in Fig. 5b. The retrieved $\xi(z_r)$ is a linearly increasing trace with a steep slope at the air-exposed region where $\xi(z_r)$ increases faster due to the larger FSBS induced local refractive index change. In the initial part of the optical fibre, where $\xi(z_r)$ is the lowest, $J_0^2(\xi(z_r))$ is the highest while $J_1^2(\xi(z_r))$ and $J_2^2(\xi(z_r))$ are the lowest (see Fig. 3b). This indicates that the SNR for $\xi(z_r)$ deduced from Eq. 8 is dominated by $J_1^2(\xi(z_r))$ and thus the SNR at the beginning of the retrieved $\xi(z_r)$ is always low (Fig. 5b).

The local response of FSBS is represented by $a_{RI}(\nu_F, z)$. From Eq. 4, $a_{RI}(\nu_F, z)$ is determined from the differentiation of the phase factor $\xi(z_r)$. To mitigate the impact of the measurement noise on the numerical differentiation, the process begins by segmenting the data points of $\xi(z_r)$ into lengths of $\Delta z$ and calculating the average value $\langle\xi(z_r)\rangle$ for the respective segment. $\Delta z$ defines the final spatial resolution, which is set as 15 m in this experiment to achieve a good balance between the number of resolved points and the SNR. Based on Eq. 4, $a_{RI}(\nu_F, z)$ can be retrieved via numerical differentiation:

$$a_{RI}(z) = \frac{\lambda}{2\pi} \frac{\langle\xi(z_r)\rangle - \langle\xi(z_r - \Delta z)\rangle}{\Delta z}$$

(9)

The retrieved $a_{RI}(\nu_F, z)$ is shown in Fig. 5c. The noise source of $a_{RI}(\nu_F, z)$ is mainly attributed to the inhomogeneity of the BGS peak amplitude that could arise from several factors including strain variations along the optical fibre due to coiling and an imperfect compensation of the BSBS probe light polarisation fading in the traces.

**Demonstration of local acoustic impedance sensing.** Local acoustic impedance sensing is demonstrated by placing the 30 m uncoated optical fibre segment in three different outer environments (air, ethanol and water). The distributed FSBS spectra of the 730 m sensing fibre with air, ethanol, and water exposures are shown in Fig. 6. The exposed region could be clearly identified in the FSBS distributed spectra maps because of the strong resonance from the efficient transverse acoustic wave reflection at the fibre boundary especially when exposed to air. The rest of the sensing fibre keeps its original acrylate coating, which reduces significantly the acoustic wave reflection. The local FSBS spectral response in the coated segments can be nevertheless observed, though with a much reduced amplitude contrast.

The FSBS resonance spectra at the exposed fibre segment are shown in Fig. 7. In the presence of different outer media, the FSBS resonance linewidth changes in an inverse proportion to the resonance peak intensity; i.e. as the resonance linewidth broadens, the resonance peak intensity decreases. The FSBS spectrum of the normal acrylate-coated fibre segment (Fig. 7, cyan) are plotted alongside as reference for comparison. The FSBS resonance linewidth of each liquid sample[17] is related to the acoustic impedances $Z_o$ of the outer material and $Z_f$ of the fibre material by

$$\delta\nu_m - \delta\nu_s = \frac{\Delta\nu_m}{\pi} \ln\left(\left|\frac{Z_o + Z_f}{Z_o - Z_f}\right|\right)$$

(10)

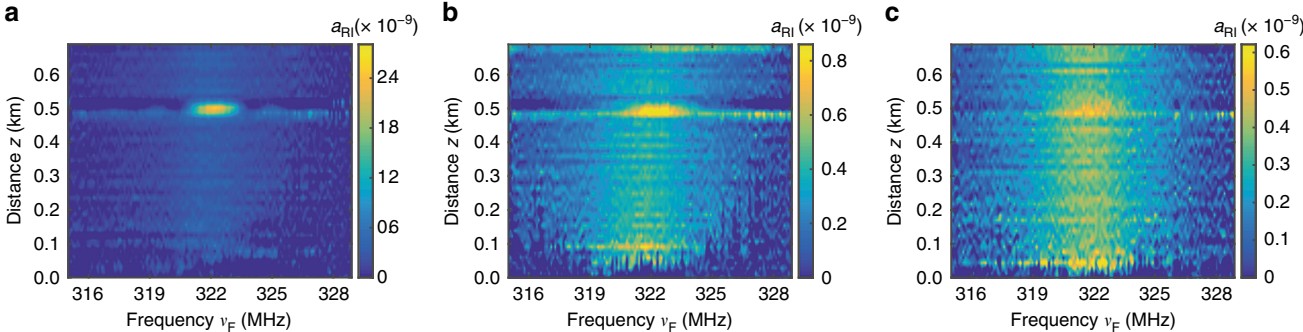

**Fig. 6** Results of the distributed measurements of forward stimulated Brillouin scattering. Distributed spectra of forward stimulated Brillouin scattering (FSBS) measured over the 730 m-long sensing fibre when the 30 m uncoated fibre segment is exposed to **a** air, **b** ethanol, and **c** water. The FSBS responses represented by the local amplitudes of refractive index changes $a_{RI}(z)$ could be clearly observed in the exposed fibre regions (0.5 km)

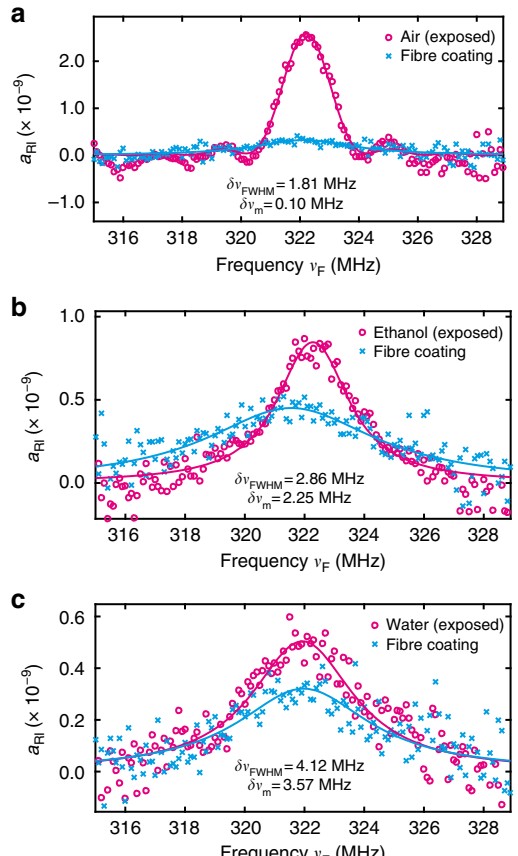

**Fig. 7** Spectra of forward stimulated Brillouin scattering at the sensing point. The spectra of forward stimulated Brillouin scattering (FSBS) measured at a given fibre position inside the 30 m fibre segment (magenta) exposed to **a** air, **b** ethanol, and **c** water. The FSBS spectra measured at an arbitrary position in the normal coated fibre segment (with acrylate polymer coating) are plotted in cyan. $\delta\nu_{FWHM}$ is the full width at half maximum (FWHM) of the measured spectrum at the sensing point (convolution of the pulse spectrum and FSBS resonance spectrum) and $\delta\nu_m$ is the recovered original linewidth of FSBS resonance at the corresponding position

where $\delta\nu_m$ is the measured FSBS linewidth, $\delta\nu_s$ is the intrinsic FSBS linewidth of the silica fibre, and $\Delta\nu_m = 48$ MHz is the average frequency separation between the chosen $\nu_{res} = 322$ MHz and the two adjacent FSBS spectral resonances[9]. $Z_f$ is given by the acoustic impedance[24] of silica ($13.1 \times 10^6$ kg m$^{-2}$ s$^{-1}$).

The intrinsic acoustic loss defining $\delta\nu_s$ is estimated from the resonance linewidth when the uncoated fibre section is exposed to air, since under this condition nearly perfect acoustic reflection at the optical fibre boundary is obtained owing to the large acoustic impedance mismatch. The intrinsic linewidth $\delta\nu_s$ is considered in this case as solely due to the bulk viscous damping of silica[20,25] (~100 kHz). The measured full width at half maximum (FWHM) $\delta\nu_{FWHM}$ of the FSBS resonance spectrum for air, ethanol, and water are respectively 1.81 MHz, 2.86 MHz, and 4.12 MHz. However, because of the finite duration of the FSBS activating pulse, the measured FSBS spectrum is a convolution of the pulse spectrum (sinc function) and the original FSBS resonance spectrum (Lorentzian function). Therefore, a full analytical expression of the broadened gain spectrum for a given activating pulse duration[26] is used here to fit the measured FSBS spectrum. The original FSBS resonance linewidth $\delta\nu_m$ successfully retrieved from the fittings are 0.10 MHz, 2.25 MHz, and 3.57 MHz for air, ethanol, and water exposures, respectively. For the case of air, the spectral response is nearly entirely given by the pulse spectrum. Since according to references[20,25] the intrinsic linewidth due to bulk viscous damping of silica $\delta\nu_s$ is about 0.1 MHz and this value is compatible with our retrieved $\delta\nu_s$ from the measurement result, we shall reasonably use it for the rest of our calculations. The corresponding acoustic impedances of the outer materials at the exposed fibre segment are then calculated based on Eq. 10, resulting in $1.49 \times 10^6$ kg m$^{-2}$ s$^{-1}$ for water, and $0.93 \times 10^6$ kg m$^{-2}$ s$^{-1}$ for ethanol. These results agree well with the reported standard acoustic impedance values[24] of water: $1.483 \times 10^6$ kg m$^{-2}$ s$^{-1}$, and ethanol: $0.93 \times 10^6$ kg m$^{-2}$ s$^{-1}$.

## Discussion

We demonstrate a technique to measure the distributed FSBS spectrum and retrieve the local acoustic impedance of the surrounding materials. In particular, the acoustic impedances of water and ethanol obtained by this technique agree well with the reported standard values. Experimental results prove that the distributed acoustic impedance measurements of a material are possible using the FSBS opto-acoustic interaction without direct interaction between light and the external material, which brings crucial advantages in terms of reliability and ease of implementation. The achieved spatial resolution $\Delta z = 15$ m is determined by the length of averaging segment in numerical differentiation. It is set to balance between the number of resolved points and the noise of differentiation. Further developments aiming at reducing the

noise sources, in particular the amplitude fluctuation of BGS peak along the fibre, could improve the spatial resolution down to the reading pulse width.

The stripping of the fibre acrylate coating in this work is to simplify the acoustic impedance analysis so that the focus of this work - distributed FSBS spectrum measurement could be emphasised. This technique operates with similar performance using optical fibres coated with a thin layer of polymer, e.g. polyimide. Thin polymer coating allows the transverse acoustic waves to penetrate with lower acoustic damping while maintains the same mechanical strength of fibre as the standard coated fibre. To demonstrate the feasibility of using thin polymer coated fibre for sensing, we have experimentally measure the acoustic impedance of ethanol and water using a commercial 80 μm-diameter SMF coated with 8 μm-thickness polyimide layer, the results are presented in Supplementary Note 2 and Supplementary Figures 3 & 4.

In addition, this technique could be deployed to characterise local FSBS spectrum inside a waveguide, thus having potential applications not only in sensing but also in quality control of hybrid photonic–phononic waveguide fabrication, where the overall acoustic gain response is sensitive to small variations in the local waveguide geometry[27,28]. Even though the present technique is clearing the way by proving the feasibility for distributed measurements, a margin for improvement in terms of noise reduction remains. The countless sophisticated possibilities to retrieve distributed information along an optical fibre make the authors confident that improved configurations will soon emerge and they believe that this proposed technique is a starting point opening new class of position-resolved sensors by taking full advantage of the distributed sensing potentiality of optical fibres.

## Methods

**Brillouin optical time domain analysis**. Brillouin optical time domain analysis (BOTDA) is a time-of-flight measurement technique that maps the resonance spectrum of SBS along an optical fibre. In the standard scheme, the technique involves using an optical pump pulse to interact with a CW counter-propagating probe light that is instrumentally red-shifted by a frequency that is within the variation range of BFS of an optical fibre. By detuning this pump-probe frequency offset around the BFS, the local Lorentzian-shaped BGS can be reconstructed. Under small gain approximation, the optical power at the central frequency of the local BGS, $P_c$, can be expressed as[2]:

$$P_c(z) = g_B P_p(z) \Delta z \tag{11}$$

where $g_B$ is Brillouin gain coefficient and $\Delta z$ is spatial resolution. Since $g_B$ and $\Delta z$ are parameters independent of $z$ for the same type of fibre and duration of pump pulse, $P_c(z)$ is therefore linearly proportional to the peak power of pump pulse at each fibre position. The full temporal trace of the probe represents the power evolution of the propagating pump pulse. This local light intensity interrogation technique based on BOTDA has previously been used to monitor 4-wave mixing signals with the aim of obtaining the position-resolved value of chromatic dispersion[21,22].

**Data availability**. The data that support the findings of this study are available from the corresponding author on request.

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

## Acknowledgements

This research is funded by Swiss National Science Foundation (200021L_157132)

## Author contributions

All authors contributed to the idea and the conceptual design. D. M. C. built the experiments, performed the measurements, analysed the data, wrote the manuscript and prepared the figures. Z. Y. contributed to the noise analyses. M. A. S. contributed to Brillouin optical time domain analysis. L. T. contributed to the light phase recovery and supervised the project. Z.Y., M.A.S. and L.T. revised the manuscript and made insightful comments.
