## [Peer Review File · Nature Communications]

Editorial Note: This manuscript has been previously reviewed at another journal that is not operating a transparent peer review scheme. This document only contains reviewer comments and rebuttal letters for versions considered at Nature Communications. Mentions of the other journal have been redacted.

REVIEWERS' COMMENTS:

Reviewer #2 (Remarks to the Author):

The authors revised their original paper and improved it by far in terms of clarity. I appreciate the new concept figures and the revised version of the explanation of the concept. The authors made a real effort to address the critics and comments of all reviewers. Following the suggestion of Reviewer 3 and dividing the description of the process into 4 parts is also a very good idea.

I am still convinced that this work is of high technical quality and now that is better understandable for a broader readership it might have enough impact for a broader community. They also clarified the practicability of their concept with coated fibers.

I have only one comment:

The authors might like to consider changing the title to something that includes an indication about the fact that their paper focuses on a distributed measurement of forward Brillouin scattering. Otherwise other researchers might not directly find in this context.

After this great revision effort, I recommend it for publication in Nature Communications.

Reviewer #6 (Remarks to the Author):

The revised version of the manuscript is significantly improved compared to the version previously submitted to [redacted]. In my opinion, the authors have convincingly addressed all reviewers concerns and have implemented the requested revisions. The new figures really help understanding the basic concept and improved the manuscript.

In general, the manuscript is well-written and all the experimental data are very well presented. There is no doubt that new distributed fiber sensors architectures have attracted significant attention recently as they can open up commercial and scientific possibilities.

I find the current manuscript quite novel and valuable to the scientific community. Their fiber sensor has been thoroughly characterized and the simulated and experimental results are in excellent agreement. From technical point of view, I cannot see any weak point of the manuscript. Therefore, I definitely recommend the publication of the current manuscript.